# Comparative functional genomics identifies unique molecular features of EPSCs

Vikas Malik[1],* , Ruge Zang[1],*, Alejandro Fuentes-Iglesias[2], Xin Huang[1], Dan Li[1], Miguel Fidalgo[2], Hongwei Zhou[1], Jianlong Wang[1]

**Extended pluripotent or expanded potential stem cells (EPSCs) possess superior developmental potential to embryonic stem cells (ESCs). However, the molecular underpinning of EPSC maintenance in vitro is not well defined. We comparatively studied transcriptome, chromatin accessibility, active histone modification marks, and relative proteomes of ESCs and the two well-established EPSC lines to probe the molecular foundation underlying EPSC developmental potential. Despite some overlapping transcriptomic and chromatin accessibility features, we defined sets of molecular signatures that distinguish EPSCs from ESCs in transcriptional and translational regulation as well as metabolic control. Interestingly, EPSCs show similar reliance on pluripotency factors Oct4, Sox2, and Nanog for self-renewal as ESCs. Our study provides a rich resource for dissecting the regulatory network that governs the developmental potency of EPSCs and exploring alternative strategies to capture totipotent stem cells in culture.**

## Introduction

Totipotent cells can give rise to the whole conceptus, including embryonic and extraembryonic tissues, whereas pluripotent cells can only give rise to embryonic tissues. Thus, totipotent cells have a superior developmental potential over pluripotent cells. However, despite the derivation of mouse embryonic stem cells (ESCs) under stable culture with maintained pluripotency more than four decades ago (Evans & Kaufman, 1981; Martin, 1981), the capture of totipotent cells in stable culture is challenging. The pioneering efforts from the Liu and Deng groups led to the in vitro stabilization of expanded potential stem cells ("L-EPSCs" hereafter) (Yang et al, 2017a) and extended pluripotent stem cells ("D-EPSCs" hereafter) (Yang et al, 2017b), collectively known as EPSCs, resembling the earlier cleavage stages of embryonic development.

EPSCs have been derived from various sources, including four- or eight-cell mouse embryos, human fibroblast-derived induced pluripotent stem cells (iPSCs), mouse and human ESCs (Yang et al, 2017a, 2017b), pig (Gao et al, 2019), and bovine (Zhao et al, 2021) early blastocysts. These cells can be further adapted to grow in feeder-free (Zheng et al, 2021) and xeno-free (Liu et al, 2021) conditions, thus opening new avenues for their molecular dissection and clinical applications. Compared with ESCs, EPSCs display superior developmental potential as they can generate both embryonic and extraembryonic tissues, including yolk sac and placenta (Yang et al, 2017a, 2017b). Furthermore, EPSCs can directly give rise to ESCs, trophoblast stem cells (TSCs), and extra-embryonic endoderm (XEN) cells using proper defined culture conditions (Yang et al, 2017a).

Although the developmental potential of EPSCs relative to their in vivo counterparts is still challenged (Posfai et al, 2021b), these cells have been nonetheless proven valuable with multiple benefits over ESCs. For example, EPSCs show superior directed differentiation potential to generate functional hepatocytes transcriptionally closer to the primary human hepatocytes than hESC-derived counterparts (Wang et al, 2020). Compared with ESCs, EPSCs can be easily derived from a non-permissive humanized mouse model (NOD-SCID Il2rg−/− strain [Du et al, 2019]). Attributing to its higher proliferation rate and better genetic and epigenetic stability, a single EPSC can give rise to an entire mouse via tetraploid complementation (Yang et al, 2017b). In addition, mouse EPSCs have superior developmental potency and robust germline competence compared with conventional ESCs, allowing for the rapid generation of gene-targeted EPSC-derived mouse models through tetraploid complementation (Li et al, 2019a). EPSCs of both mouse and human origins outperform pluripotent stem cells (PSCs) in interspecies chimera in mouse conceptuses (Yang et al, 2017b; Gao et al, 2019) and monkey embryos cultured ex vivo (Tan et al, 2021). Similarly, EPSCs combined with TSCs (Sozen et al, 2019) or EPSCs alone (even a single EPSC) could form blastoids, blastocyst-like structures that can further develop to post-implantation embryo structure and

[1]Department of Medicine, Columbia Center for Human Development, Columbia Stem Cell Initiative, Herbert Irving Comprehensive Cancer Center, Columbia University Irving Medical Center, New York, NY, USA   [2]Department of Physiology, Center for Research in Molecular Medicine and Chronic Diseases (CiMUS), Universidade de Santiago de Compostela (USC)-Health Research Institute (IDIS), Santiago de Compostela, Spain

Correspondence: jw3925@cumc.columbia.edu
*Vikas Malik and Ruge Zang contributed equally to this work.

 

induce decidualization in vitro, although they fail to generate live pups (Li et al, 2019b). Therefore, further studies to understand and improve their developmental potency are much needed.

Previous studies reported the epigenome and single-cell transcriptome of EPSCs (Yang et al, 2017a, 2017b; Posfai et al, 2021b); however, a precise molecular makeup of EPSCs is still missing. Here, we reprogrammed mouse ESCs to D-EPSCs and L-EPSCs using the respective protocols (Yang et al, 2017a, 2017b) and systematically mapped transcriptome, chromatin accessibility, active enhancer and promoter marks, and proteomes of D/L-EPSCs relative to ESCs. We found that, despite similar reliance on key pluripotency factors Oct4, Sox2, and Nanog for their maintenance, EPSCs differ from ESCs in molecular features including expression of other pluripotency-associated (e.g., *Lin28a*, *Utf1*, *Esrrb*, *Nr5a2*, and *Myc*), DNA methylation (e.g., *Dnmt3a/b/l* and *Mettl4*), and gastrulation (e.g., *Eomes*, *Dusp4*, *Bmp4*, and *Lef1*) related genes. We also uncovered differentially open chromatin genomic loci harboring DNA motifs of RAR-RXR and Zfp281 in L- and D-EPSCs, respectively. In addition, our proteomics data revealed the differences in specific translational and metabolic regulation in ESCs, D-EPSCs, and L-EPSCs. Together, our study provides a rich resource for further dissecting the regulatory network governing the unique developmental potential of EPSCs.

# Results

### Transcriptome comparison reveals discernible gene expression changes between EPSCs and ESCs

To dissect the molecular features of EPSCs, we first converted ESCs (cultured in 2i/leukemia inhibitory factor [LIF] medium) to D-EPSCs and L-EPSCs (Fig 1A) following previously published protocols (Yang et al, 2017a, 2017b). We observed compact EPSC colonies with smooth edges with and without a feeder layer (Fig 1B), with L-EPSCs forming slightly flat colonies without feeder layers, consistent with a previous study (Posfai et al, 2021b). Next, we performed bulk RNA-seq to examine how a transcriptome shift is induced after switching ESCs into either EPSC conditions. Replicates correlated well, and the transcriptomic profiles of both EPSC lines were closer to each other than to the ESCs (Fig 1C), and principal component analysis (PCA) showed global gene expression variability in the three cell states (Fig 1D). Differential gene expression analysis revealed that the transcriptomes of ESCs show much larger gene expression differences with D-EPSCs (1,875 up-regulated and 2,024 down-regulated genes) and L-EPSCs (2,128 up-regulated and 1,619 down-regulated genes) than those between L-EPSCs and D-EPSCs (836 up-regulated in L-EPSCs and 1,573 up-regulated in D-EPSCs) (Fig 1E and Table S1), consistent with the correlation heat map (Fig 1C). The expression levels of Oct4 and Sox2 in both EPSCs resembled those in ESCs at mRNA (Figs 1E and F and S1A) and protein levels (Fig 1G), although Nanog showed a slightly lower mRNA level in EPSCs relative to ESCs and yet similar protein levels in EPSCs and ESCs (Fig 1E–G). EPSCs also showed reduced expression of a few pluripotency genes, including *Nr5a2* and *Esrrb*, while overexpressing other pluripotency-

associated genes such as *Utf1*, *Lin28a*, *Dnmt3l*, *Zic3*, and *Myc* (Figs 1F and S1A). Interestingly, whereas most early totipotent two-cell specific genes (i.e., 2C markers) do not express in ESCs or EPSCs, some of them, including *Zscan4c/d/f* and *Usp17le*, albeit lowly expressed, express at slightly higher levels, especially in L-EPSCs (Figs 1F and S1A), with an enrichment of H3K27ac active histone modification mark near their promoter regions (Fig S1B) than ESCs.

The intersection of differentially expressed genes (DEGs) in ESCs versus EPSCs from this study and previously published bulk RNA-seq of ESCs versus day 15 EPSCs (Posfai et al, 2021b) showed that over half (>55%) DEGs (734/1,334) in the latter study were recapitulated in our studies (Fig S1C). However, we captured a larger number of DEGs that distinguish D/L-EPSCs from ESCs, possibly due to different sequencing platform and depth as well as inherent transcriptomic heterogeneity among different ESC lines from different mouse strains. GSEA analysis using DEGs from all the three comparisons (Fig 1E) showed enrichment for cell fate commitment and embryonic development in ESCs and both EPSC lines (Fig 1H). Interestingly, D-EPSCs showed enrichment of FGF signaling pathway, whereas L-EPSCs are enriched for gastrulation-related terms. Consistently, both EPSC lines showed a strong enrichment of DNA methylation signature and a significant increase in expression levels of DNA methylation-associated genes (n = 22), including *Dnmt3a/b/l* and *Mettl4* (Figs 1I and S1A). In contrast, only L-EPSCs showed a significantly higher expression of gastrulation-related genes (n = 29) (Figs 1I and S1A).

In sum, these data indicate that, whereas the expression of Oct4, Sox2, and Nanog genes is similar in ESCs and EPSCs, some pluripotency and totipotency-related genes are differentially overexpressed in EPSCs relative to ESCs. Thus, EPSCs do reflect a departure from ESCs at the transcriptome level in both pluripotency and totipotency-related gene expression.

### Chromatin accessibility comparison identifies a subset of putative transcriptional regulators for the unique developmental potential of EPSCs

To examine how the gene expression differences observed above would be correlated with the chromatin status between ESCs and EPSCs, we probed chromatin accessibility by ATAC-seq (assay for transposase-accessible chromatin using sequencing) (Buenrostro et al, 2013, 2015). Our data are highly concordant between replicates (Fig S2A), and PCA revealed the occurrence of accessibility differences in the three cell populations (Fig S2B). Differential accessibility analysis of ATAC-seq peaks showed a large proportion of shared open chromatin among ESCs and both EPSC lines (Fig S2C), yet there also occur subsets of genomic regions that showed significant accessibility changes (Fig 2A). Differential accessibility analysis yielded six groups from three comparisons, that is, (i) ESCs (E) versus D-EPSCs (D): sites open (O) in ESCs and closed (C) in D-EPSCs (EO-DC n = 1,753) and vice versa (DO-EC n = 9,113); (ii) ESCs versus L-EPSCs (L) (EO-LC n = 525; LO-EC n = 2,670); and (iii) L-EPSCs versus D-EPSCs (LO-DC n = 798; DO-LC n = 1,193) (Fig 2A and Table S2). These comparisons reveal that ESCs show more differences in their chromatin opening with D-EPSCs than L-EPSCs (Figs 2A and S2A). The ATAC-seq signals corresponded to the chromatin accessibility group classification according to the identified six groups (Fig 2A)

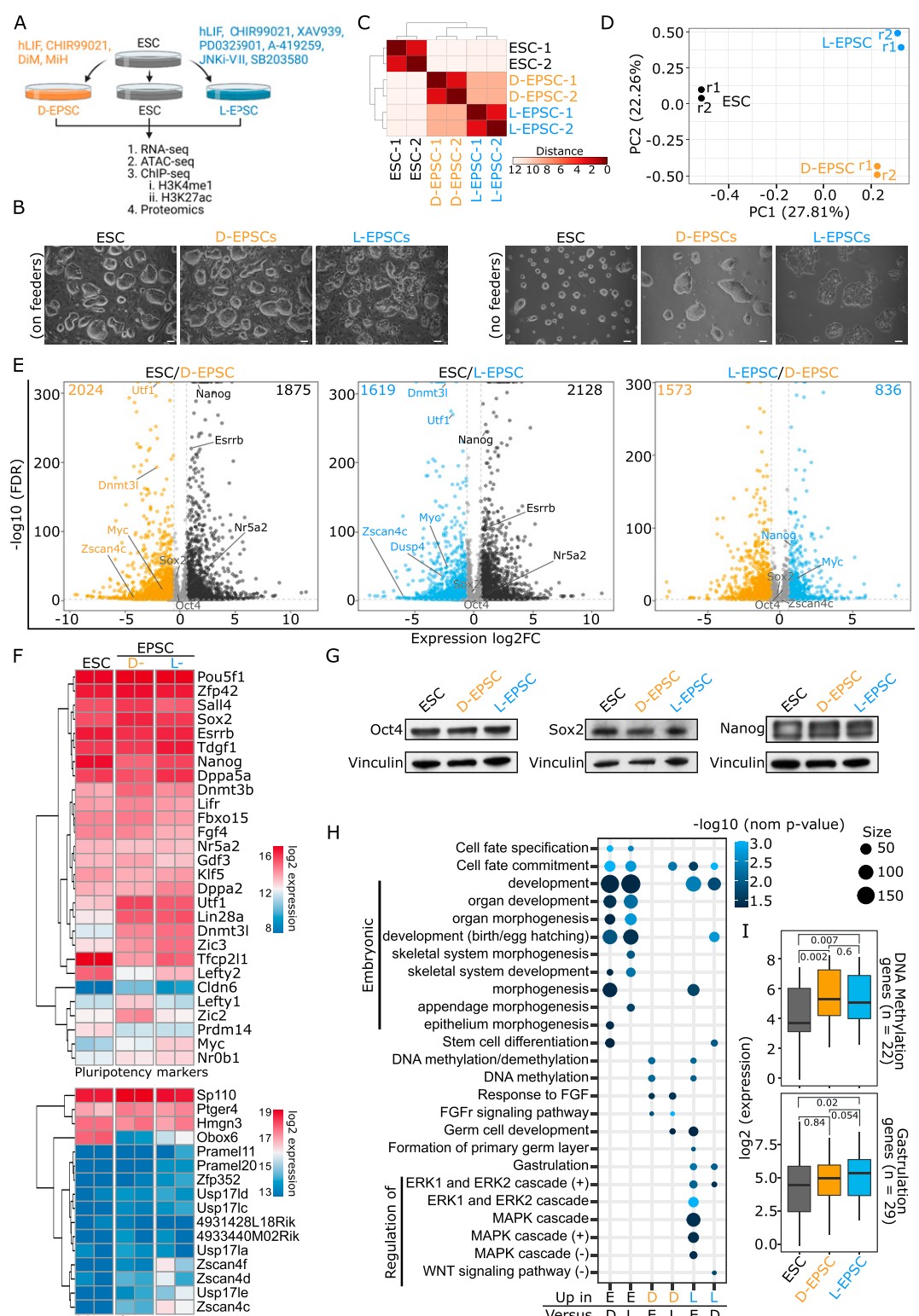

**Figure 1. Expanded potential stem cells (EPSCs) share transcriptomic profiles with embryonic stem cells (ESCs) but with discernible differences in pluripotency and 2C genes.**
**(A)** A summary of the experimental design. **(B)** Representative phase-contrast images of indicated cellular states with (left panels) or without (right panels) feeders at passage 7. The scale bar is 100 μm. **(C)** Hierarchical clustering correlation heat map of the RNA-seq signals from ESC, D-, and L-EPSCs replicates. Distance, calculated with DEseq2 (Love et al, 2014), represents sample-to-sample distances. **(D)** Principal component analysis for the RNA-seq gene expression data. **(E)** Volcano plots of differential gene expression between ESC and D-EPSC (left panel), ESC and L-EPSC (middle), and L- and D-EPSC (right) with selected candidates highlighted. The horizontal

with representative loci shown (Fig 2B). These analyses suggest that, despite major genome-wide chromatin accessibility overlap among ESCs and both EPSC lines (Fig S2C), there exist subsets of differentially accessible regions between EPSCs and ESCs and between the two EPSC types.

The genomic distribution of differentially open peaks in D/L-EPSC lines compared with ESCs showed a stronger enrichment around the TSS region, including CpG island, 5′ UTR, and promoter regions (Fig 2C), indicating that access to gene-proximal regulatory elements may play a significant role during the ESC-to-EPSC conversion. Furthermore, in comparison to ESCs, the differentially open ATAC-seq peaks containing LTRs were depleted in both EPSC lines (Fig 2C), which corresponds with significantly lower expression levels of genes associated with these LTR-containing regions in EPSCs than ESCs (Fig 2D). Similar results were reported by a previous study showing reduced expression of porcine endogenous retroviruses (PERVs) in porcine-EPSCs (Kruger et al, 2021), suggesting a cross-species conserved phenomenon.

To gain mechanistic insight into the unique chromatin dynamics of EPSCs, we analyzed the DNA motifs associated with the overlapping and differentially accessible peaks in ESCs and both EPSC lines. The overlapping peaks among all three cell types mainly harbored the pluripotency-related TF motifs (Fig S2D), whereas the differentially accessible six groups (defined in Fig 2A) showed the presence of various classes of TF motifs in addition to the pluripotency related motifs (Fig 2E and Table S2). We further filtered the motifs and only kept the ones (n = 12) present in more than 50% of open peaks in ESCs, D-EPSCs, and L-EPSCs (Figs 2F and S2E). We found that ESCs are enriched for Tcf/Lef1-like motifs, D-EPSCs for Ronin/Gfy-Staf, REST-NRSF, Six4, NRF, and Zfp281 motifs, whereas L-EPSCs for retinoic acid receptor-retinoid X receptor (RAR-RXR) heterodimer binding motifs (Figs 2F and S2E). The enrichment of the Zfp281 motif in D-EPSCs and the RAR-RXR motif in L-EPSCs is contrasting and intriguing because Zfp281 inhibits (Wen et al, 2022), whereas the retinoic acid (RA)-signaling pathway promotes the transition of ESCs to totipotent two-cell–like cells (2CLCs) (Iturbide et al, 2021). Nonetheless, we noticed that the promoter regions of Zfp281 together with Rara, Rarg, Rxrb, and Rxrg genes among RAR and RXR gene families are accessible and enriched for active histone marks. Consistently, we found that their transcripts are overexpressed in EPSCs compared with ESCs (Fig S2F).

In sum, compared with ESCs, EPSCs show enrichment of a unique subset of TF motifs, including Zfp281 and RAR-RXR with corresponding gene expression changes and closing of LTR containing regions and their reduced gene expression, which may provide critical regulatory elements that drive the expanded potential of EPSCs, a hypothesis that warrants future investigations.

## Histone marks, transcription, and chromatin accessibility based molecular features of EPSCs

To investigate the potential impact of differential open chromatin (Fig 2A) on differential gene expression in EPSCs compared with ESCs, we performed ChIP-seq of H3K4me1 and H3K27ac to characterize active promoters/enhancers in ESCs and EPSCs and expand the resource for these cells in addition to the previously characterized bivalent marks (H3K4me3 and H3K27me3) in ESCs (Bernstein et al, 2006) and EPSCs (Yang et al, 2017a). Both ChIP-seq replicates correlate well (Fig S3A), enabling us to identify the chromatin environment and signature genes regulating EPSCs fate with the following approaches. First, we defined the peaks with differential enrichment of histone marks in ESCs versus D/L-EPSC lines (Fig S3B and C). Second, we performed a combinatorial analysis by intersecting the regions (i) with differential enrichment of histone marks (Fig S3B), (ii) that are accessible (Fig 2A), and (iii) with their nearby genes up-regulated in both EPSCs and ESCs (Fig 1E). As a result, we defined 131 and 85 signature genes for EPSCs and ESCs, respectively (Fig 3A and Table S3). Third, we performed GO analysis of these identified signature genes, revealing DNA methylation (e.g., *Dnmt3a*), pluripotency network (e.g., *Tfap2c*, *Zmym2*, and *Lefty1*), and MAPK/ERK signaling pathways (e.g., *Rara*, *Fgfr2*, and *Dusp6*) are prominent features associated with EPSC signature genes, whereas otic vesicle development (e.g., *Fgf10*, *Fgf3*, and *Eya1*) and LIF signaling pathways (e.g., *Nr5a2*, *Hk2*, *Trim2*, *Kat6b*, and *Arid5b*) are more pronounced in ESC signature genes (Fig 3B). The representative candidates from 131 EPSC signature genes have open promoter regions, higher H3K4me1 and H3K27ac signals, and higher mRNA expression levels in either D- or L-EPSCs compared with ESCs (Fig 3C). Of note, Tfap2c is a trophoblast (Kuckenberg et al, 2010) and naïve pluripotency marker (Pastor et al, 2018). Among 272 mouse cell lines or tissue samples (Hutchins et al, 2017), Tfap2c transcripts are expressed at high levels in placenta, trophoblast stem cells, 4/8-cell embryos followed by ESCs (Fig S3D). We noticed that, unlike its mRNA expression (Fig 3C), Tfap2c is expressed at higher protein levels in both D/L-EPSCs than ESCs (Fig S3E), suggesting a potential posttranscriptional regulation. To understand its functional role in EPSCs, we used Tfap2a/c double knockout (dKO) mouse ESCs (Pastor et al, 2018). We found that under respective culture medium, Tfap2a/c dKO ESCs could be efficiently converted to D-EPSCs but less so to L-EPSCs compared with WT ESCs, evident with morphological changes (flat colonies with individualized cells) as signs of differentiation already within the first five passages which could not be further passaged and maintained (Fig S3F). These results indicate the differential requirement of Tfap2a/c for the EPSC state by different culture conditions, suggesting D- and L-EPSCs may represent two distinct expanded pluripotency states.

and vertical dashed lines represent the false discovery rate (FDR, 0.05) and log$_2$ fold change (±0.6) cut-offs, respectively. The numbers of differentially expressed genes are indicated on the top corners and details provided in Table S1. **(F)** Heat maps show the expression dynamics of selected pluripotency (top) and 2C (bottom) markers genes for ESC, D-EPSC and L-EPSC. Data are presented as log$_2$ normalized counts. **(G)** Western blots for Oct4, Sox2, and Nanog in three cell types. Vinculin is the loading control. **(H)** GSEA of the differentially expressed genes in E filtered by FDR < 0.05, log$_2$ FC > 0.6 & < −0.6. Select GO terms were extracted using keywords "Embryonic, FGF, Gastrulation, Stem cell, MAP, ERK, DNA methylation" out of 927 significant terms from six groups. Size represents the number of genes associated with each term and color −log$_{10}$ nominal *P*-value. **(H, I)** Box plots of RNA-seq signals of selected genes associated with GSEA terms DNA methylation (n = 22, Table S1) and gastrulation (n = 29, Table S1) from heat map in (H). Indicated *P*-values were calculated using the unpaired Mann–Whitney U test also known as Wilcoxon rank-sum test (R function Wilcoxon test).

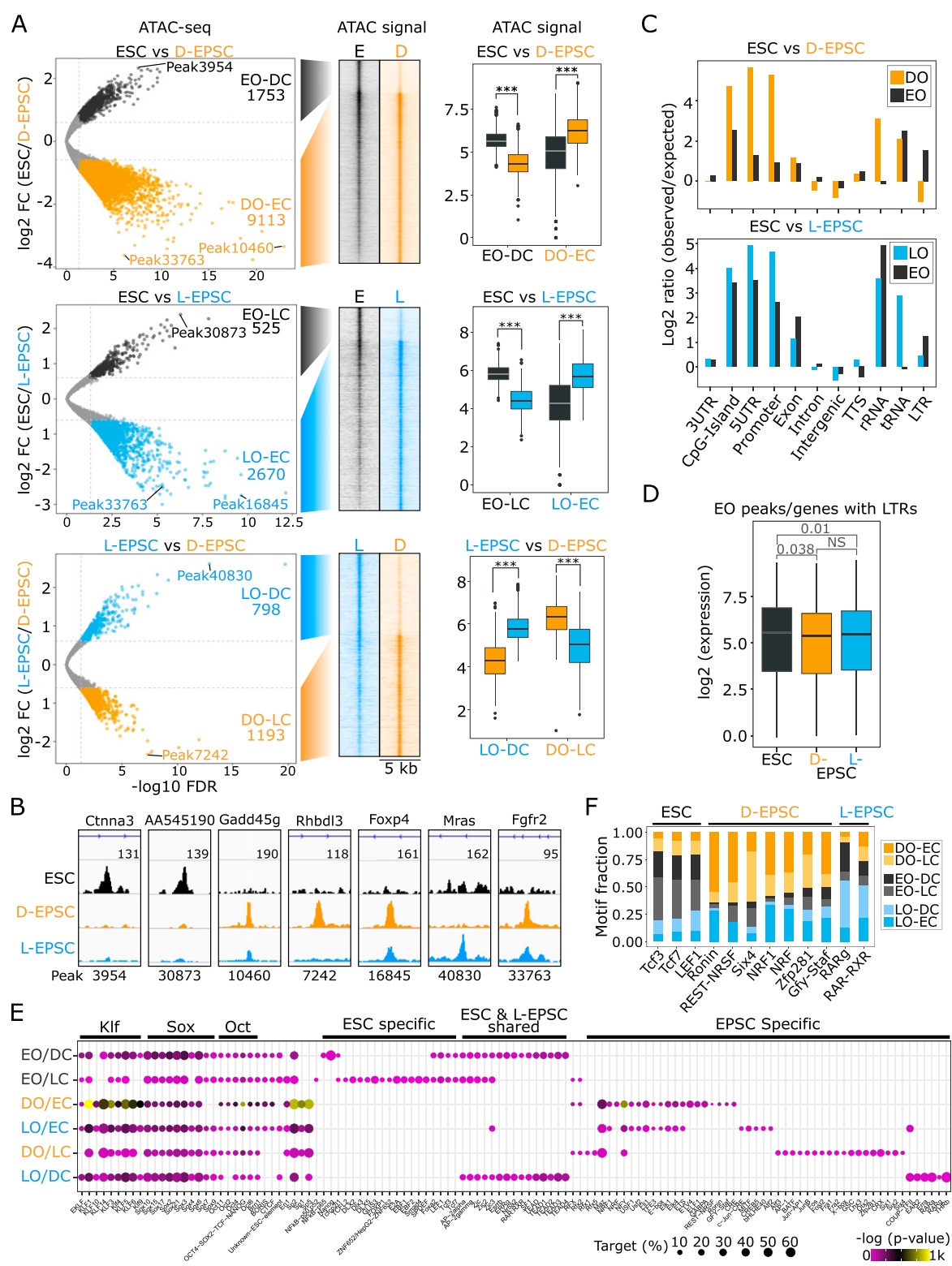

**Figure 2. Expanded potential stem cells (EPSCs) show differential chromatin accessibility near the promoter region than embryonic stem cells (ESCs).**
**(A)** Volcano plots of differential genomic accessibility performed using DiffBind tool (Ross-Innes et al, 2012) between ESC and D-EPSC (top panel), ESC and L-EPSC (middle), and L- and D-EPSC (bottom); named using acronyms: Open (O) and Close (C) in ESC (E), D-EPSC (D), and L-EPSC (L) conditions. Peak numbers and select peaks with differential accessibility in each category are indicated. A heat map of the ATAC-seq signal (RPM normalized) within a 2.5-kb window centered at six accessibility groups was drawn using EaSeq (Lerdrup et al, 2016) and corresponding boxplots. *P*-values (*P < 0.05; **P < 0.01; ***P < 0.001) were calculated using the unpaired Mann–Whitney Wilcoxon Test (R function Wilcox test). Genomic locations are provided in Table S2. **(A, B)** Representative genome browser tracks of selected ATAC-seq

The overexpression of *Zmym2* and *Rara* in EPSCs is intriguing because we and others have shown that Zmym2 and RAR family proteins play negative (Yang et al, 2020) and positive (Iturbide et al, 2021) roles, respectively, in 2CLC totipotency induction. Nonetheless, the enrichment of RAR-RXR motif in more than 50% differential open region in L-EPSCs (Fig 2F) and the overexpression of RAR and RXR family members in both EPSCs (Fig S2F) suggest the potential roles of RAR/RXR factors in EPSC fate regulation. Collectively, we identified a subset of potential EPSC signature genes that could serve as prime candidates in constructing the regulatory network governing the unique developmental potential of EPSCs.

### Like ESCs, EPSCs also rely on Oct4 and Sox2 but not Nanog for their maintenance

Our data indicate that although we identified EPSC-specific gene signatures, EPSCs also share with ESCs some of the gene expression, chromatin accessibility, and histone marks. In particular, we wondered to what extent EPSCs are dependent on the pluripotency network for their maintenance. In this regard, we focused on the core pluripotency factors Oct4 (Niwa et al, 2000), Sox2 (Masui et al, 2007), and Nanog (Chambers et al, 2003; Das et al, 2011). We used previously established ZHBTc4, 2TS22C, and NgcKO [tet-off]ESC lines for tetracycline/doxycycline-induced conditional knockout (cKO) of Oct4 (Niwa et al, 2000), Sox2 (Masui et al, 2007), and Nanog (Das et al, 2011), respectively. In these ESC lines, the respective tet-off transgene (Fig 4A) sustains the self-renewal of [tet-off]ESCs that are genetically null for each endogenous gene (Fig 4A) before the doxycycline (Dox) treatment. We were thus able to convert them to stable [tet-off]L-EPSCs without Dox (Fig 4A and B) using the published protocol (Yang et al, 2017a). By adding Dox to turn off the Oct4 and Sox2 transgenes, we observed the collapse of EPSCs concomitant with the protein loss (Fig 4B and C). In contrast, despite the Nanog protein loss (Fig 4C), Nanog-cKO [tet-off]ESCs (NgcKO), and their converted [tet-off]L-EPSCs could still be maintained, although with reduced size and number of colonies (Fig 4B), indicating a similar role of Nanog in L-EPSC maintenance as that of ESC maintenance. Of note, it is well known that ESCs without Nanog can be maintained although they proliferate slower and are more prone to differentiation (Chambers et al, 2007).

To understand the downstream effect of the loss of Oct4, Sox2, and Nanog proteins on EPSCs relative to ESCs, we performed RNA-seq in all three [tet-off]ESC lines and corresponding [tet-off]L-EPSC lines with and without Dox. All replicates correlated well with each other (Fig S4A). All three [tet-off]ESC lines are separated from their individually derived [tet-off]L-EPSC lines at the transcriptome level (Fig 4D). However, whereas the Dox-treated Oct4-cKO and Sox2-cKO [tet-off]L-EPSCs clustered separately from their untreated counterparts, Dox-treated and untreated Nanog-cKO [tet-off]L-EPSCs

clustered together (Figs 4D and S4A). These data indicate two major gene expression changes happen, first during the establishment phase, that is, [tet-off]ESC-to-[tet-off]L-EPSC conversion in all three cKO cell types, and second during the maintenance phase, that is, Oct4-cKO and Sox2-cKO [tet-off]L-EPSCs after Dox addition. Of note, there were only minimal or modest gene expression changes in Nanog-cKO [tet-off]L-EPSCs upon Dox addition, consistent with the minimal effect of the Nanog loss on EPSC morphology/maintenance (Fig 4B). Consistently, global DEG analysis in Dox-treated and untreated [tet-off]L-EPSCs revealed that both Oct4-cKO and Sox2-cKO [tet-off]L-EPSCs showed more up-regulated (n = 1,517 and n = 1,407) and down-regulated (n = 1,895 and n = 1,261) genes compared with only a few hundred (up-regulated = 202 and down-regulated = 304) in Nanog-cKO [tet-off]L-EPSCs (Fig 4E and Table S4). GO analysis of these differentially regulated, especially down-regulated, genes in all three conditions revealed pluripotency network as one of the most affected terms (Fig S4B). The expression of major pluripotency-related markers was significantly down-regulated in Oct4-cKO and Sox2-cKO but not in Nanog-cKO [tet-off]L-EPSCs (Figs 4F and S4C). In contrast, the expression of totipotency-associated 2C marker genes did not show a significant change in all three cell lines (Fig S4D).

Puzzling enough, we could not generate [tet-off]D-EPSCs from these cKO cell lines despite repeated trials following strictly the published protocol (Yang et al, 2017b) (indicated by a red "X" in Fig 4A). We suspected that was due to the similarity of structure and mechanism of action between Dox and minocycline hydrochloride (MiH), a chemical compound required for the induction of D-EPSCs (Yang et al, 2017b). To ascertain this is the case, we resorted to an Oct4-degron ESC system allowing for rapid Oct4 protein degradation with dTAG treatment (Boija et al, 2018). Indeed, we could successfully convert Oct4-degron ESCs to both [tet-off]D- and [tet-off]L-EPSCs without dTAG treatment (Fig S4E), and dTAG addition induced Oct4 protein degradation concomitant with differentiation of [tet-off]ESCs and both [tet-off]EPSC lines and the eventual collapse of all these cells. Collectively, our data indicate that like ESCs, EPSC maintenance is also critically dependent on pluripotency factors Oct4 and Sox2 and, to a much lesser extent, Nanog.

### The proteomic comparison reveals differential translational and metabolic control between ESCs and EPSCs

To understand the functional outcomes of the global genomic and chromatin differences in EPSCs versus ESCs, we interrogated the differential proteome in ESCs and EPSCs. We performed quantitative proteomics using SILAC-based MS with biological replicates to achieve a high-accuracy analysis of the proteome for each cellular state (Fig 5A). We identified a total of 1,103 proteins among which 129 and 79 are up-regulated ($\log_2$ SILAC ratios > 0.6 & < −0.6) in

---

peaks (indicated in A). The nearest genes are displayed on the top, peak number identifiers at the bottom, and signal strength is shown with respective numbers. **(C)** Genomic distribution of ATAC-seq peaks from indicated groups are shown as bar plots. **(D)** Boxplots for RNA-seq signals of nearest genes associated with ATAC-seq peaks harboring LTRs from combined-ESC-open peaks compared with D- & L-EPSC categories. Indicated *P*-values were calculated using the unpaired Mann–Whitney Wilcoxon Test (R function Wilcox test); NS, not significant. **(E)** Enrichment of selected motifs in six differential-ATAC-seq categories obtained using HOMER (Heinz et al, 2010). Point size represents the proportion of sequences with the motif, and color gradient the *P*-value score. Major motif classes are indicted on top; list provided in Table S2. **(E, F)** Proportion of ATAC-seq peaks featuring select top motifs (from E) in each accessibility category. Motifs from (E) were selected if they occurred in more than 50% of ATAC-seq peaks in EO, DO, and LO categories.

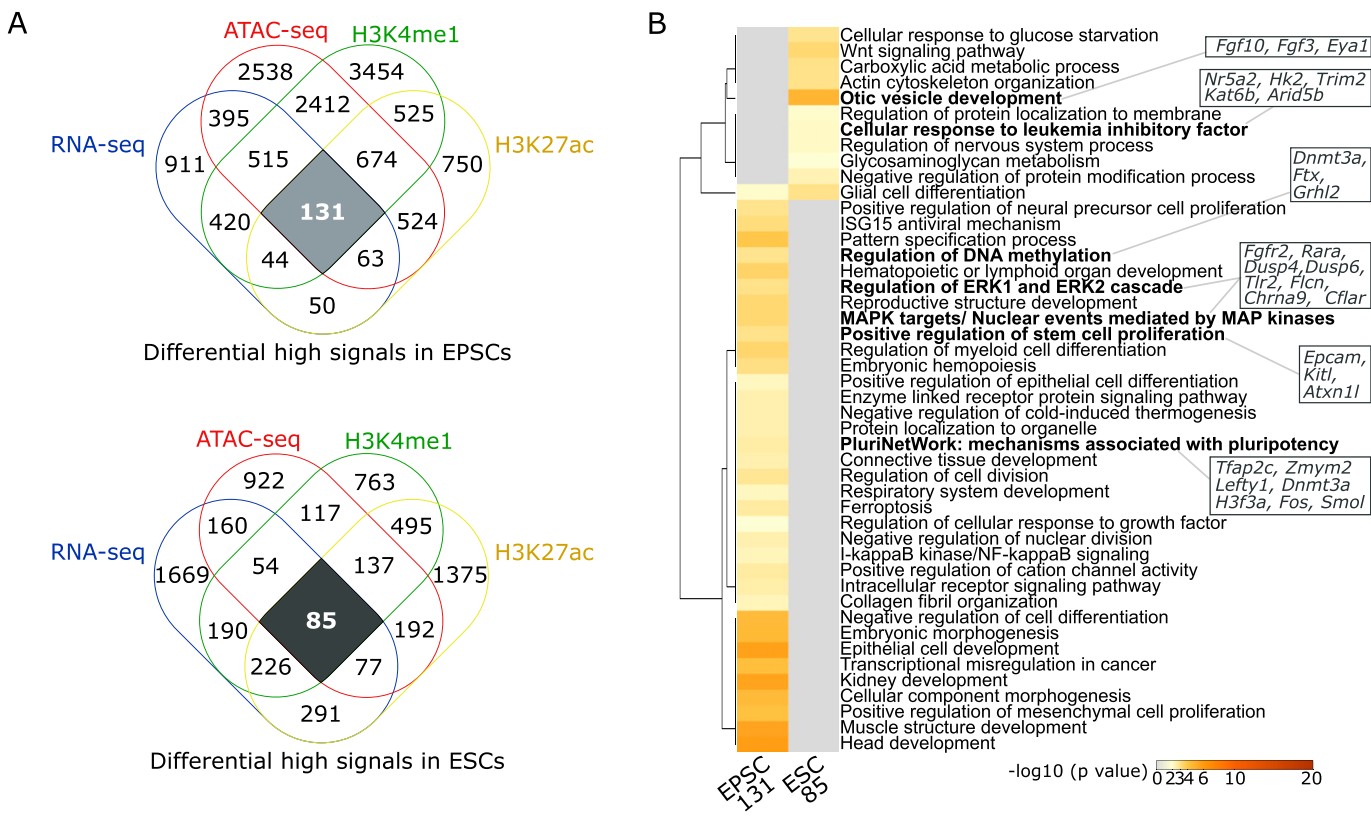

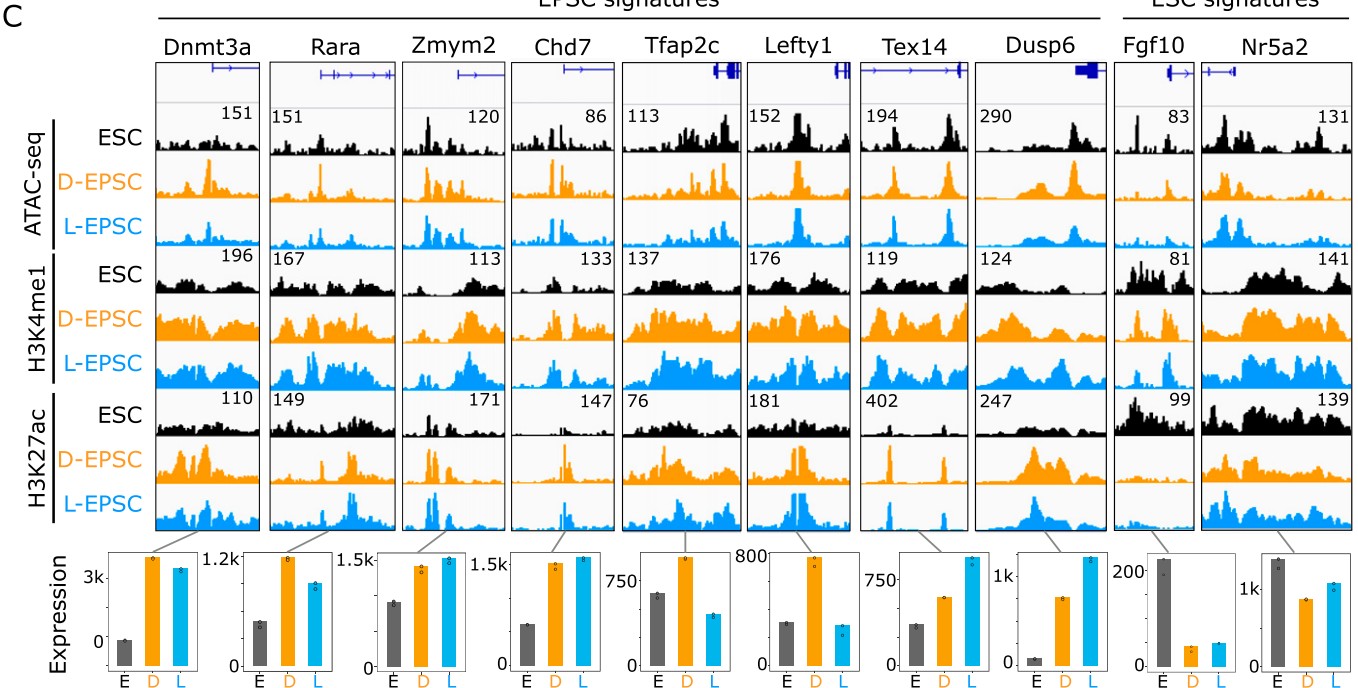

**Figure 3. Defining the molecular signature of expanded potential stem cells (EPSCs).**
**(A)** Venn diagram showing the intersection of overexpressed genes from RNA-seq, open regions from ATAC-seq with the presence of H3K4me1 and H3K27ac signal in EPSCs (top) and embryonic stem cells (bottom). **(A, B)** GO analysis of 131 and 85 genes from (A) using metascape (Zhou et al, 2019). Genes associated with selected terms are indicated. **(A, C)** Genome browser tracks represent ATAC-seq and indicated histone marks ChIP-seq signals around promoter regions of selected genes from 131 or 81 candidates in (A) and their expression levels in three cell states. ATAC-seq and ChIP-seq signal strengths, set same for embryonic stem cell, D-EPSC, L-EPSC samples, are indicated once with respective numbers for each gene loci.

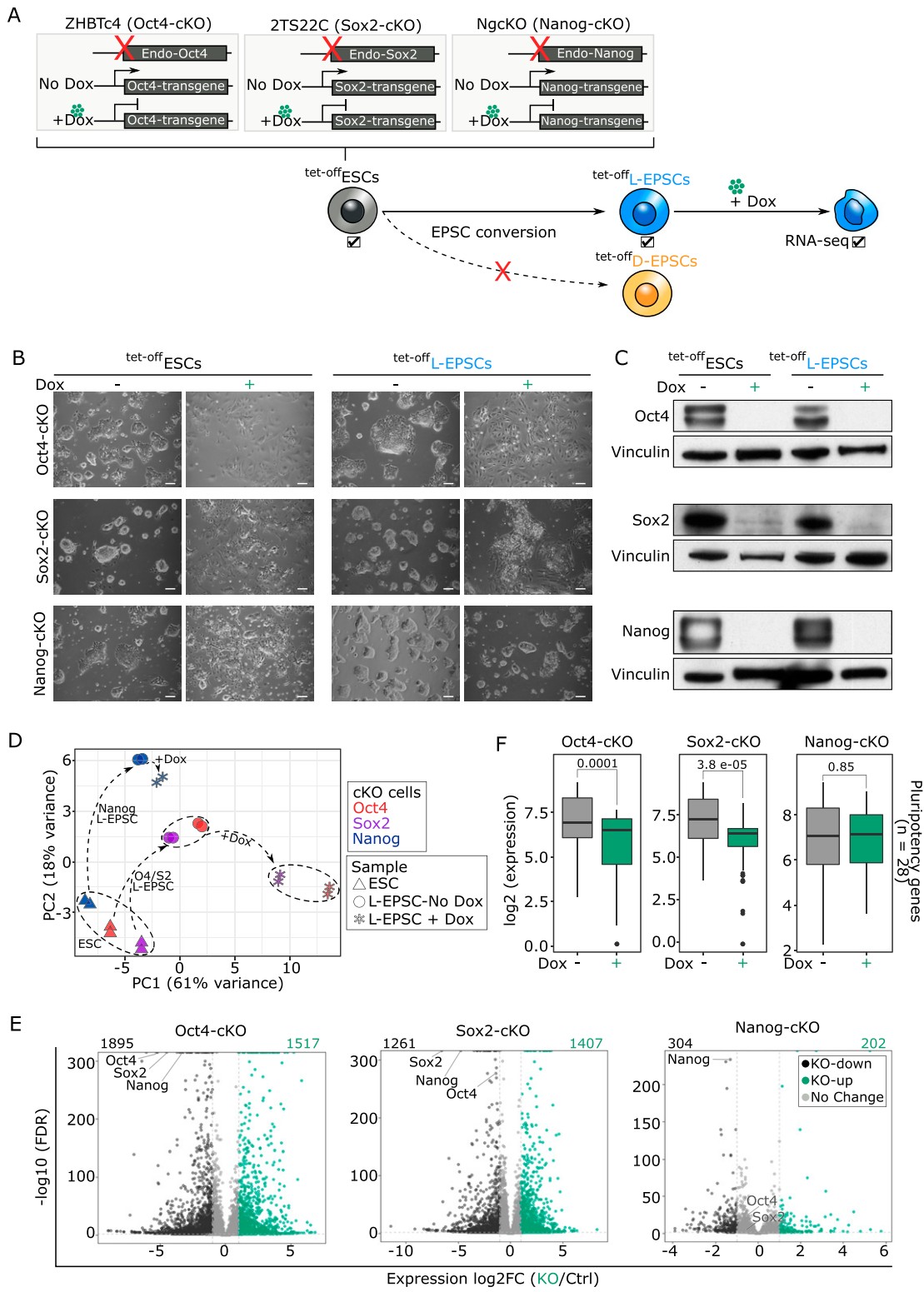

**Figure 4. Requirement of Oct4, Sox2, and, to a lesser extent, Nanog for expanded potential stem cell (EPSC) maintenance.**
**(A)** Illustration of RNA-seq experiment (tick marks) on [tet-off]L-EPSCs converted from *Oct4* (Niwa et al, 2000), *Sox2* (Masui et al, 2007), and *Nanog* (Das et al, 2011) conditional knockout (Oct4-, Sox2-, and Nanog-cKO) [tet-off]ESCs, wherein each of these endogenous (endo) genes is deleted, and cells are maintained by their corresponding doxycycline (Dox)-suppressible transgene. **(B, C)** Cellular morphology (B) and protein expression (C) on day 5 after dox treatment (1 μg/ml) is shown in [tet-off]ESCs and [tet-off]L-EPSCs. The scale bar is 100 μm. Vinculin is the housekeeping control. **(D)** Principal component analysis for the RNA-seq gene expression from indicated samples. **(E)** Volcano plots of differential gene expression of Dox-treated versus untreated [tet-off]L-EPSCs from O/S/N-cKO backgrounds. Differentially expressed

ESCs compared with D-EPSCs and L-EPSCs, respectively (Fig 5B). As expected, we observed higher levels of Parp1 in ESCs than both L- and D-EPSCs (Fig 5B) because both chemical Parp1 inhibition and genetic Parp1-KO are reported to be beneficial for EPSC maintenance and developmental potency (Yang et al, 2017b). D-EPSCs overexpress pluripotency-related proteins like Lin28a and Dnmt3l, whereas Utf1 is overexpressed in both D- and L-EPSCs compared with ESCs (Figs 5B and S5A). The GO analysis of proteins overexpressed in D-EPSCs compared with ESCs showed enrichment of terms related to posttranscriptional gene regulation and translation (Fig 5C). Consistently, D-EPSCs showed enrichment of eukaryotic translation initiation factors, namely, Eif3b/c/e/m, Eif4g1, Eif6 and translation initiation-associated ribosomal protein Rps15a compared with ESCs (Figs 5B and S6). During mouse early embryonic development, the transcript levels of the above-mentioned translation initiation factors are higher at E5.5 (an embryonic stage that correlates with D-EPSCs at the single-cell level [Posfai et al, 2021b]) than ESCs (Fig S5B). On the other hand, compared with D-EPSCs, L-EPSCs specifically showed enrichment of Eif4a2 protein (Fig 5B). Similarly, the transcript levels of Eif4a2 are abundant in the E4.0 state and subsequently decrease in both ESCs and further in D-EPSCs (Fig S5B). Notably, L-EPSCs are correlated at the mRNA level with the E4.5 stage (Posfai et al, 2021b). We recently showed that Eif4a2 mediates specialized translational control of mRNA targets governing stem cell and developmental potency (Li et al, 2022). Thus, overexpression of Eif4a2 in L-EPSCs compared with both ESCs and D-EPSCs and other translation-related factors in D-EPSCs suggest that the two EPSC types might use different translational mechanisms to control expanded pluripotency.

The GO analysis for proteins up-regulated in L-EPSCs compared with both ESCs and D-EPSCs showed enrichment of terms like TCA cycle and fatty acid $\beta$-oxidation (Fig 5C). We observed overexpression of multiple TCA cycle-related proteins, namely, Aco2, Sdha/b, Ndufs1, Ndufs3, Ndufa9, Vdac1, and Acad9 in L-EPSCs compared with both ESCs and D-EPSCs (Figs 5B and S6). In contrast, ESCs mainly overexpress glycolytic proteins (e.g., Hk2, Tigar, Atp1a1, Vdac3, and Pgk1) and G1/S transition related proteins (e.g., Cdk2, Cdk4, Mcm3, and Mcm4) compared with D-EPSCs and L-EPSCs, respectively. Among these proteins, the majority of TCA (e.g., Aco2, Ndufs1, Ndufs3, Ndufa9, Sdha, and Sdhb) and glycolysis (e.g., Atp1a1, Hk2, and Tigar) related factors showed concordant expression at both mRNA and protein levels in ESCs versus L-EPSCs (Fig S7). For the rest of the proteins (e.g., Acad9, Pgk1, and Vdac3) the mRNA and protein expression were not concordant suggesting possible posttranscriptional regulation. These results suggest that ESCs and EPSCs, especially L-EPSCs, likely depend on different metabolic needs, supported by previous studies demonstrating that mouse early embryos (with totipotency and/or extended pluripotency) are dependent on pyruvate (TCA cycle), whereas ESCs on glucose (glycolytic pathway) as their main energy source (Nagaraj et al, 2017; Zhang et al, 2018a).

In sum, our proteomics data on three cellular states highlight that EPSCs may use differential translational and metabolic

controls to acquire their expanded potential over ESCs, pointing another direction for future mechanistic inquiries into the EPSC biology.

## Discussion

The Deng (Yang et al, 2017b) and Liu (Yang et al, 2017a) laboratories simultaneously established EPSCs with unique developmental potential over ESCs, which ushered in valuable applications of these cells in enhanced directed differentiation (Wang et al, 2020), blastoid and interspecies chimera generation (Sozen et al, 2019; Li et al, 2019b; Tan et al, 2021), and faster mouse model generation (Li et al, 2019a). Despite the challenge raised on their expanded potential (Posfai et al, 2021b), EPSCs were subsequently further proved to be advantageous over ESCs in interspecies monkey–human chimera generation (Tan et al, 2021) and the derivation of totipotent potential stem cells (TPSCs) (Xu et al, 2022). These studies definitely highlight the untapped potential of EPSCs for further exploration to drive studies towards in vitro establishment of authentic totipotent stem cells, although the molecular foundation of these EPSCs are still poorly characterized. Here we used comprehensive genomics, epigenetic and proteomics approaches to compare the molecular features of the two EPSC lines and their similarities and differences compared with ESCs that remained unexplored by previous studies. Our data demonstrate that EPSCs express the core pluripotency factors Oct4, Sox2, and Nanog but differentially overexpress other pluripotency-associated factors such as *Utf1*, *Dnmt3a/b/l*, *Zfp281*, and *Zmym2* compared with ESCs (Fig 6). In addition, EPSCs, especially L-EPSCs, also show slight up-regulation of *Zscan4c*, a totipotency-associated factor, compared with ESCs (Figs 1F and 6). Apart from highlighting the subtle differences in pluripotency and totipotency related genes which Posfai et al (2021a, 2021b) had overlooked, we also showed how subsets of genes related to DNA methylation and gastrulation (Figures 1H and I) could be explored further to guide studies that are focused on establishing the authentic totipotent stem cells in vitro.

The actual developmental potency of totipotent-like cells generated across various laboratories remains murky because these cell types show variable molecular features (Yang et al, 2017a, 2017b; Sozen et al, 2019; Li et al, 2019b; Tan et al, 2021; Posfai et al, 2021b; Xu et al, 2022). Thus, it is critical to know where precisely these cells map/position on the early development trajectory (Posfai et al, 2021a; Malik & Wang, 2022). Single-cell transcriptomic profile comparison with mouse early embryonic development had classified L-EPSCs slightly earlier (E4.5) than D-EPSCs (E.5.5) (Posfai et al, 2021b). We also noticed some remarkable differences in a few molecular features that suggest the placement of L-EPSCs at a slightly earlier developmental stage than D-EPSCs. For example, higher expression of Lin28a protein, representative of primed pluripotency state (Zhang et al, 2016), was observed in D-EPSCs, whereas overexpression of TCA cycle-related proteins, a feature of

gene numbers are indicated, and a list is provided in Table S4. **(F)** Boxplots showing expression of pluripotency related genes (n = 29 from Fig 1F and Table S4) in Dox-treated and untreated tet-offL-EPSCs.

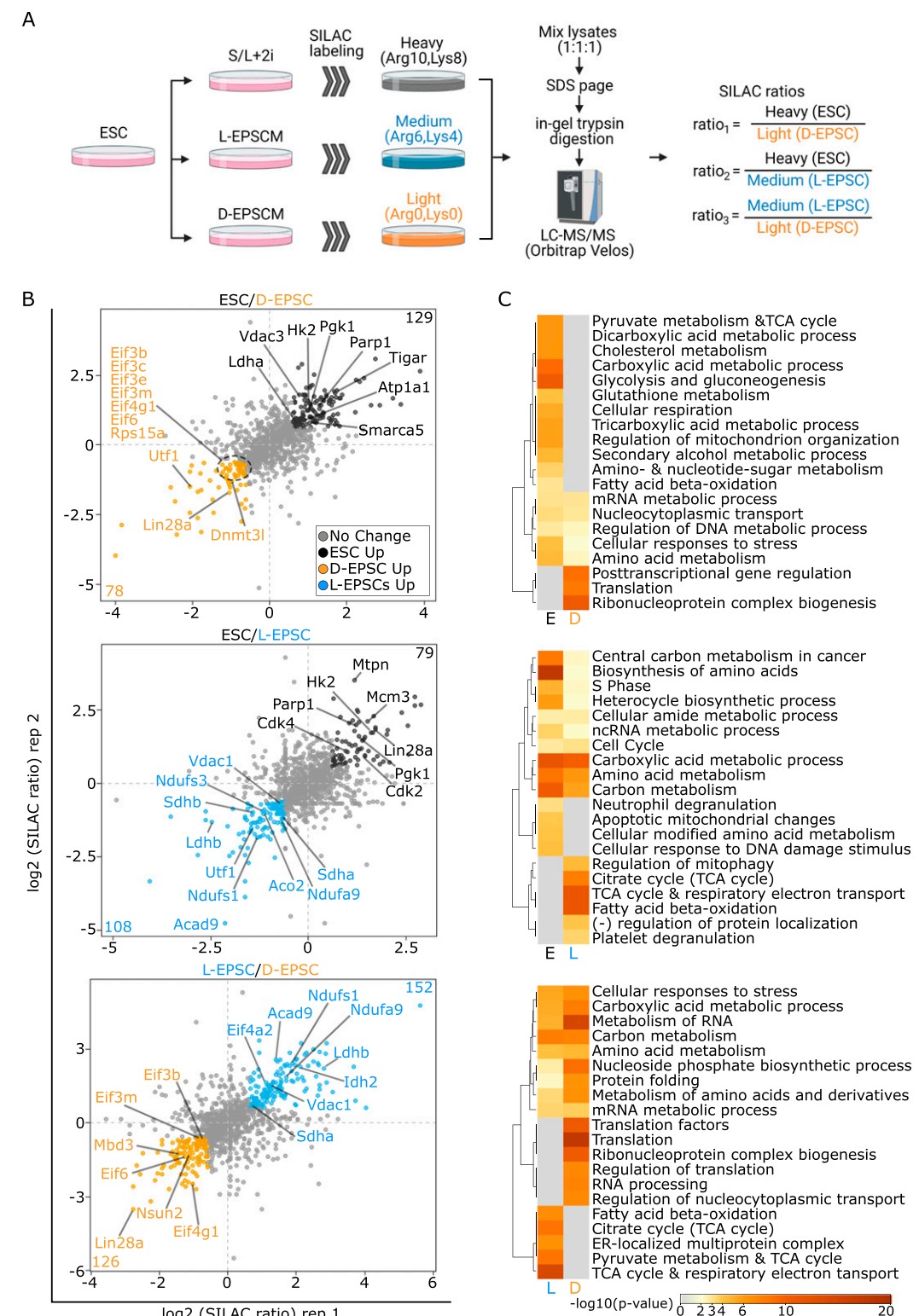

**Figure 5. D– and L–expanded potential stem cells (EPSCs) show overexpression of translation initiation and TCA cycle-related proteins.**
**(A)** Illustration showing proteomics experiment using SILAC-based mass spectrometry (MS). **(B)** Protein ratios of two independent mass spec whole proteome measurements of embryonic stem cells, D-EPSCs, and L-EPSCs SILAC labeling are shown as dot plots (Table S5). Differentially expressed proteins (log$_2$ ratio > 0.6 and < −0.6) are highlighted with darker colors, and their numbers are indicated in respective quadrants. Select proteins are labeled. **(B, C)** GO analysis using the differentially expressed proteins (log$_2$ ratio >0.6 & <−0.6 i.e., colored dots except grey) from (B).

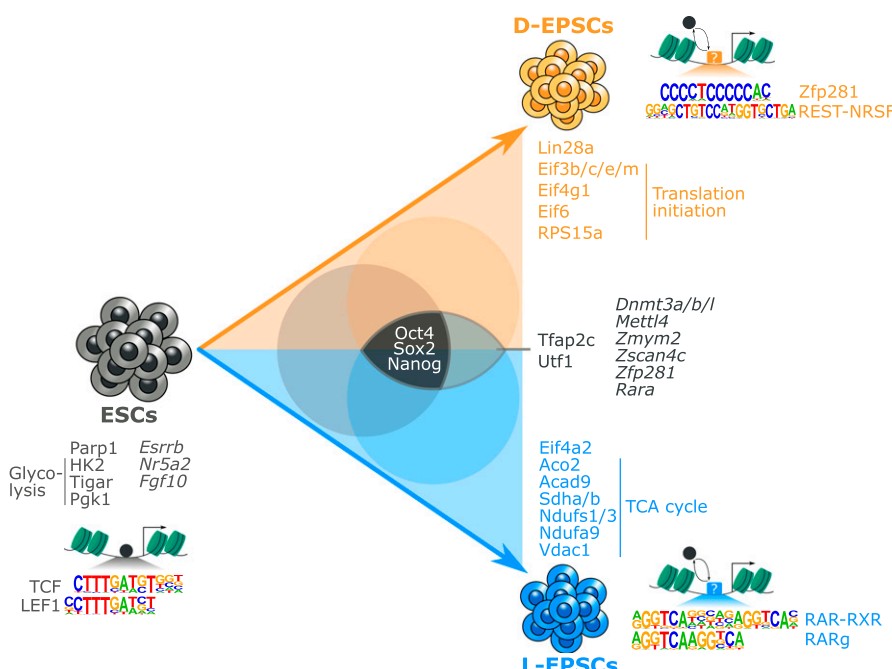

**Figure 6. Summary of the molecular features associated with embryonic stem cells, D-, and L-expanded potential stem cells (EPSCs).**
Schematic representation showing conversion of embryonic stem cells to D- and L-EPSCs with shared proteins at the center (Oct4, Sox2, and Nanog) and uniquely enriched proteins and mRNA (italics) at the edges of the triangle. D- and L-EPSCs shared proteins and mRNAs are in the middle of the two cell types. Open chromatin and enriched motifs in each cell type are indicated. The pluripotency factors (filled grey circle) and EPSCs fate regulators (squares with the question mark) are shown.

much earlier developmental stage (Nagaraj et al, 2017; Zhang et al, 2018a), was observed in L-EPSCs (Figs 5B and 6). The relatively higher proliferation rate of D-EPSCs compared with L-EPSCs could be further explored to highlight if there is any connection with their dependence on different metabolic pathways (Yang et al, 2017a, 2017b). Similarly, L-EPSCs with the overexpression of Eif4a2 are better associated with an earlier embryonic development stage, whereas D-EPSCs show overexpression of multiple translation initiation factor proteins representing a later developmental stage than ESCs. These datasets suggest an intriguing possibility that these factors may contribute to the differential developmental potential of D/L-EPSCs and/or the existence of alternative states of expanded pluripotency of EPSCs. It also raises an interesting question of whether one EPSC line might have higher plasticity than the other.

The conversion of ESCs to EPSCs led to differential chromatin opening concomitant with enrichment of various transcription factor motifs in differentially accessible sites (Figs 2E and 6). It will be interesting to know which factors are directly (pioneers) or indirectly (settlers or migrants) (Sherwood et al, 2014) involved in bringing about these chromatin accessibility changes. Immediate genome-wide occupancy studies of crucial transcription factor(s) combined with our existing ATAC-seq data analysis would help understand the dynamic processes that possibly involve passive displacement of pluripotency factors leading to chromatin closing and/or active involvement of the pioneering role of expanded pluripotency related factors leading to chromatin opening (Fig 6). In addition, it would be important to know whether major pluripotency factors, especially Oct4 and Sox2, bind to different locations in EPSCs than ESCs. Sox2 acts as a major pioneer factor in pluripotency induction (Zhu et al, 2018; Malik et al, 2019) and partners with different proteins in a cell context-dependent manner (Adachi

et al, 2013; Lodato et al, 2013), so it would be interesting to know if there are changes in Sox2 binding locations and partners in EPSCs as well. Similarly, the determination of binding locations of identified EPSC-enriched factors (Fig 6) would help us understand their regulation more precisely. *Zfp281* and *Rara* transcripts are expressed at nearly identical levels in both EPSCs (Figs S2F and 6). Yet, their DNA binding motifs are enriched differentially in the accessible regions in D- and L-EPSCs, respectively (Figs 2F and 6). To understand whether these differentially open locations containing the abovementioned DNA motifs are actually bound by their respective transcription factors, it would be of prime interest to catalog all the genome-wide locations bound by expanded potential related factors. We noticed the presence of the RAR binding motif upstream of *Eif4a2* in L-EPSCs (Fig S5C). Using our previously published ChIP-seq data (Fidalgo et al, 2016), we found that Zfp281 binds the *Dnmt3a* and *Lin28a* genomic regions in ESCs (Fig S5D). A comparative genome-wide binding study in ESCs and EPSCs will help identify more regulatory signatures like this. Such studies will thus delineate if and how Zfp281 and RAR-RXR family proteins may contribute to the expanded potential of EPSCs, particularly considering activation of RARγ signaling are important for inducing and maintaining totipotent features of TPSCs (Xu et al, 2022) and both Zfp281 and RAR-RXR family proteins may play a role in cell division and DNA replication-free reprogramming of somatic nuclei for embryonic transcription (Tomikawa et al, 2021). Recently Hu et al succeeded in capturing 2CLCs in vitro using three chemicals including TTNPB (RAR-agonist) (Hu et al, 2022), thus corroborating our findings and strengthening the demand to explore the possible role of these factors in expanded potential of EPSCs.

Open chromatin regions in the mouse genome (about 45%) are associated with repeat elements. The accessibility around these repeat elements is highly dynamic during early embryonic

development and ESCs (Lu et al, 2020). Similarly, global DNA methylation is also dynamic during mouse early development stages and controls the expression of repeat elements (Smith et al, 2012). Although it is well known that mouse early embryos and ESCs show higher expression of ERVs (regulated by LTRs), we found that, compared with ESCs, differentially open regions in EPSCs were depleted around LTRs and matched with significant down-regulation of the corresponding RNA expression (Fig 2C and D) in alignment with a previous report (Kruger et al, 2021). We also noticed that EPSCs overexpress Dnmt3 enzymes (Figs S1A, 5B, and 6). Because DNA methylation levels control the repression of repeat elements (Smith et al, 2012; He et al, 2019), it raises more questions as to whether differential overexpression of DNA methylation enzymes is at interplay to control the selective repression of LTR-containing open regions in EPSCs.

In summary, we systematically compared ESCs and EPSCs with combined genomic, transcriptomic, and proteomic approaches. We identified critical genomic loci and proteins that provide a rich resource for further investigations as the potential molecular targets endowing EPSCs with superior developmental potential over ESCs (Fig 6). Similar studies are warranted for other currently available totipotent-like cells, including 2CLCs (Macfarlan et al, 2012), totipotent blastomere-like cells (TBLCs) (Shen et al, 2021), totipotent-like stem cells (TLSCs) (Yang et al, 2022), TPSCs (Xu et al, 2022), chemically induced totipotent stem cells (ciTotiSCs) (Hu et al, 2022), and human 8CLCs (Mazid et al, 2022; Taubenschmid-Stowers et al, 2022). Ultimately, we will be able to capture authentic totipotent stem cells in vitro (Malik & Wang, 2022) and likely also reveal alternative paths to totipotency and/or alternative totipotent states.

# Materials and Methods

## ESC and EPSC culture

### Mouse ES cells (mESCs) J1
Mouse ES cells (mESCs) J1 were cultured on inactive MEF feeder cells (~30,000 cells per cm$^2$) or on 0.1% gelatin-coated plates in a serum-based medium with 2i (3 $\mu$M CHIR99021; 1 $\mu$M PD0325901) addition, which was prepared as follows: DMEM supplemented with FBS (15%), recombinant LIF (1,000 U/ml), $\beta$-mercaptoethanol (0.1 mM), L-glutamine (2 mM), MEM non-essential amino acids (NEAA, 0.1 mM), nucleoside mix (1%), and penicillin/streptomycin (50 U/ml). The medium was changed daily or every other day, and cells were passaged every 3 d using trypsin (0.05%).

### D-EPSCs
D-EPSCs were cultured in a base medium of N2B27 prepared as follows: DMEM/F12 and Neurobasal (1:1 ratio), N2 supplement (1×), B27 supplement (1×), GlutaMAX (1×), and $\beta$-mercaptoethanol (0.1 mM). The base medium was supplemented with KSR (5%), NEAA (0.1 mM), LIF (1,000 U/ml), CHIR99021 (3 $\mu$M); dimethindene maleate (DiM, 2 $\mu$M); and minocycline hydrochloride (MiH, 2 $\mu$M). L-EPSCs were cultured on feeder cells or gelatin-coated plates in a base medium prepared as follows: DMEM/F12, KnockOut Serum

Replacement (KSR, 20%), L-glutamine (2 mM), NEAA (0.1 mM), and $\beta$-mercaptoethanol (0.1 mM). The base medium was supplemented with LIF (1,000 U/ml), CHIR99021 (3 $\mu$M), PD0325901 (1 $\mu$M), A-419259 (0.3 $\mu$M), XAV939 (5 $\mu$M), JNK inhibitor VIII (4 $\mu$M), and SB203580 (10 $\mu$M). For both EPSCs, the medium was changed daily, and cells were passaged every 3 d with accutase. Both EPSCs were cultured in feeder-free conditions for about 15 d, which equated to five passages before they were used for ATAC-seq, ChIP-seq, and RNA-seq experiments.

Oct4-FKBP mESCs, used for degron (dTAG) treatment based Oct4 protein depletion experiment, were cultured on 0.1% gelatin-coated plates in N2B27/2i/LIF medium formulated as: DMEM/F12 and Neurobasal (1:1 mix) medium supplemented with N2 (1×) and B27 (1×) supplements, NEAA (0.1 mM), LIF (1,000 U/ml), CHIR99021 (3 $\mu$M), and PD03259010 (1 $\mu$M).

All the cells were cultured in an incubator with 5% $CO_2$ at 37°C. Detailed reagents and tools list is provided in Table S6.

## RNA-seq and data analysis

Total RNA from ESCs and EPSCs replicates were extracted using Trizol according to the manufacturer's protocol. RNA quality was evaluated by Agilent 2100 BioAnalyzer. Total RNA from each sample was isolated and used to prepare RNA-seq libraries. RNA-seq libraries were prepared manually using Universal Plus mRNA-Seq with NuQuant kit, according to the manufacturer's protocol. For each sample, 500 ng total RNA was used to isolate mRNA via poly(A) selection. Captured mRNA was washed, fragmented, and primed with a mix of random oligo(dT) primers. After cDNA synthesis, ends were repaired and ligated with Unique Dual Index (UDI) adaptor pairs. Libraries were amplified by 14 PCR cycles and purified with AMPure XP beads, sequenced on the NovaSeq 6000 platform with 150 bp paired-end read length with Novogene.

RNA-seq reads quality assessment and adaptor trimming of fastq files were performed using TrimGalore v.0.6.4, retaining reads with a minimum length of 60 and a minimum Phred score of 20. Processed reads were mapped against the mouse genome (mm9/NCBIM37.67) and sorted by coordinate using STAR v2.7.9a. The number of reads per gene was counted using htseq v.0.11.2, providing the genome annotation (GTF format) from the NCBIM37.67 mouse genome. Differential gene expression was analyzed using DESeq2 v.4.1.1 R package. Genes not expressed in all samples (rowSums ≤ 1) were filtered out from the analysis. Correlation plots and PCA were performed on vst-transformed values (variance stabilized transformation; implemented in DESeq2 package). Genes were considered differentially expressed if they had an FDR value of < 0.05 and a log$_2$ fold-change > ±0.6 unless otherwise indicated.

Gene Set Enrichment Analysis (GSEA v.4.2.2) was used to assess the ontology terms enriched in each sample using the C5 Gene Ontology Gene Set Database (v.7.5.1). Only significant genes (FDR < 0.05 and Log$_2$FC > ±0.6) were selected for this analysis. Briefly, P-values were calculated based on 1,000 permutations, with permutation type set to *gene_set*. Enrichment analysis was set to *weighted* for the enrichment score calculation, and *log2_Ratio_of_Classes* was used for gene ranking. Only GO terms with NOM P-val < 0.05 were retained for further analysis. Gene

Ontology terms heat map, volcano plots, PCA, and correlation plots were plotted using R software.

## ATAC-seq and data analysis

The ATAC-seq libraries of ESCs and both EPSCs were prepared in technical replicates as previously described (Buenrostro et al, 2013, 2015) with minor modification. Briefly, $5 \times 10^4$ cells were lysed by lysis buffer (10 mM Tris–HCl [pH 7.4], 10 Mm NaCl, 3 mM MgCl$_2$, and 0.15% NP-40) for 10 min on ice to prepare the nuclei. Immediately after lysis, nuclei were spun down at 500$g$ for 5 min. Next, the pellet was incubated with the Tn5 transposase and tagmentation buffer at 37°C for 30 min (Vazyme Biotech). After the tagmentation, the stop buffer was added directly into the reaction to end the tagmentation. PCR was performed to amplify the library for 15 cycles using the following PCR conditions: 72°C for 3 min; 98°C for 30 s; and thermocycling at 98°C for 15 s, 60°C for 30 s, and 72°C for 3 min; following by 72°C for 5 min. After the PCR reaction, libraries from 200 to 700 bp were purified using gel extraction before sequencing. NextGen sequencing was performed by NovaSeq 6000 platform with 150 paired-end reads.

ATAC-seq reads were processed as previously described (Yang et al, 2020). Briefly, sequencing reads were aligned to mouse genome (mm9) using the bowtie2 (v2.3.5) program, with parameters -X 2000–no-mixed. Aligned reads were filtered by samtools (v1.10) program with parameters -F 0x04 -f 0x02 -q 20. ATAC-seq peaks were determined by the MACS program (v.2.2.7) with default settings. Differential peak accessibility was determined using the Diffbind tool v.3.4.11 (Ross-Innes et al, 2012). Briefly, reads were counted over each peak, normalized, and differential analysis was performed using DESeq2. All peaks (th = 1) were extracted for further analysis. Peaks were considered differentially accessible if they had a value of FDR < 0.05 and a log$_2$ fold-change > 0.6 and < −0.6. Peak filtering and downstream analysis were performed using bedtools software v.2.27.1, including peak intersection and coverage depth computing (bedtools intersect and bedtools coverage functions, respectively). Motif analysis was performed using HOMER v.4.11.1 (findMotifsGenome.pl script). Briefly, the top 15 most significant known motifs (ranked by FDR value) were selected in all comparisons and merged. Motifs were then separated into several groups based on their sample enrichment and shown in Figs 2E and S2D. Peak genomic distribution was assessed by homer annotatePeaks.pl script (-annStats option). Log$_2$ (obs/exp) ratio was plotted in Fig 2C. Motif fraction enrichment in each sample was computed using HOMER with annotatePeaks.pl function. Briefly, the identified 88 motif types (Fig 2E) that present differential accessibility between mESCs and EPSCs were downloaded from the homer motif database (http://homer.ucsd.edu/homer/motif/HomerMotifDB/homerResults.html) and merged into a single file. A merged motif file was provided as input to HOMER using the annotatePeaks function (-m option). Motif counts were calculated, and the motif fraction was obtained by normalizing with the peak numbers in each accessibility group. The most enriched motifs

in mESC and both EPSCs were plotted in Fig 2F. ATAC-seq peak annotation was performed using the homer annotatePeaks.pl function to assign the nearest gene name to the peaks. Enrichment heat maps in Figs 2A, S2C, and S3C were plotted using EaSeq v.1.111, ranked by read coverage. Correlation and volcano plots were plotted using R software.

## ChIP-seq and data analysis

H3K4me1 and H3K27ac ChIP-seq experiments were performed in replicates as described (Ding et al, 2015). One million cells were used for each sample. Massively parallel sequencing was performed with the Illumina NovaSeq 6000 according to the manufacturer's protocol, and pair-end 150 bp length reads were produced. FastQC was used to check the sequencing quality. ChIP-seq reads were aligned to the mouse genome mm9 using the bowtie2 (v2.3.5) program, with parameters -X 1000 –no-mixed –no-discordant. The mapped reads were sorted and converted to a binary bam file using SAMTools (v1.10). ChIP-seq peaks were determined by the MACS program (v.2.2.7) with the -broad option, using input as the control data. Differential peak enrichment was determined using Diffbind software (similar as in the "ATAC-seq and data analysis" section). Peak annotation was performed using HOMER annotatePeaks.pl script to assign nearby genes to each peak.

## SILAC-MS

The SILAC-MS procedure is illustrated in Fig 4A. Briefly, D-EPSCs were cultured in SILAC Light (Lys0, Arg0), L-EPSCs were cultured in SILAC Medium (Lys4, Arg6), and ESCs were cultured in SILAC Heavy (Lys8, Arg10) media for at least five passages. Cell lysates of each population were equally mixed for the following steps. Protein lysates were dissolved in 8M Urea buffer, followed by in-gel tryptic digestion and liquid chromatography-tandem mass spectrometry (LC–MS/MS) using an Orbitrap-Velos mass spectrometer. Proteome Discoverer Software with mouse proteome was used for protein quantification and identification. The relative intensities of heavy, medium, and light fraction for each protein were exported for further analysis.

## Western blot

Whole-cell protein extracts were isolated from the cultured cells using RIPA lysis buffer (NC9193720; Boston BioProducts) supplemented with protease inhibitor cocktail (P8340; Sigma-Aldrich) and phosphatase inhibitor cocktail (78428; Thermo Fisher Scientific). Blots were incubated in 2% BSA/TBST at room temperature for 1 h, and then they were incubated with the corresponding antibodies in 5% skimmed milk powder/TBST at 4°C overnight. Secondary antibodies were anti-rabbit IgG, HRP-linked antibody (1:5,000; R&D System, HAF008) and anti-mouse IgG, HRP-linked antibody (1:5,000; Cell Signaling Technology, 7076S), which were incubated for 1 h at room temperature while shaking. The blots were developed using XRAY FILM (Cat. no. XAR ALF 2025; LabScientific) in a dark room.

## Data Availability

Sequencing data (ATAC-seq, ChIP-seq, and RNA-seq) that supports the findings of this study are available in Gene Expression Omnibus. Accession number for the study is GSE201305.

## Supplementary Information

## Acknowledgements

We thank Drs. Amander T Clark, Richard A Young, Austin Smith, and Hitoshi Niwa for providing Tfap2a/c dKO, Oct4-degron ESCs, ZHBTC4 ESCs, and 2TS22C ESCs, respectively. This work in the Wang laboratory is funded by grants from the National Institutes of Health (R01HD095938 and R01HD097268) and by contracts from New York State Stem Cell Science (NYSTEM#C35583GG). A Fuentes-Iglesias is the recipient of a fellowship from the MINECO of Spain (BES-2017-082007).

### Author Contributions

V Malik: data curation, formal analysis, validation, investigation, methodology, and writing—original draft, review, and editing.
R Zang: data curation, investigation, methodology, and writing—review and editing.
A Fuentes-Iglesias: data curation, formal analysis, investigation, and writing—review and editing.
X Huang: data curation, formal analysis, investigation, methodology, and writing—review and editing.
D Li: data curation, investigation, methodology, and writing—review and editing.
M Fidalgo: writing—review and editing.
H Zhou: data curation, investigation, methodology, project administration, and writing—review and editing.
J Wang: conceptualization, supervision, funding acquisition, investigation, visualization, methodology, project administration, and writing—original draft, review, and editing.

### Conflict of Interest Statement

The authors declare that they have no conflict of interest.

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
