## [Reviewer comments · Life Science Alliance]

Life Science Alliance

Comparative functional genomics identifies unique molecular features of EPSCs

Vikas Malik, Ruge Zang, Alejandro Fuentes-Iglesias, Xin Huang, Dan Li, Miguel Fidalgo, Hongwei Zhou, and Jianlong Wang
DOI: <https://doi.org/10.26508/lsa.202201608>

Corresponding author(s): Jianlong Wang, Columbia University Medical Center

Review Timeline:

Submission Date:	2022-07-15
Editorial Decision:	2022-07-15
Revision Received:	2022-07-15
Editorial Decision:	2022-07-19
Revision Received:	2022-07-19
Accepted:	2022-07-20

Transaction Report:

Please note that the manuscript was previously reviewed at another journal and the reports were taken into account in the decision-making process at Life Science Alliance. Since the original reviews are not subject to Life Science Alliance's transparent review process policy, the reports and author response cannot be published.

July 15, 2022

Re: Life Science Alliance manuscript #LSA-2022-01608-T

Jianlong Wang
Columbia University Irving Medical Center

Dear Dr. Wang,

Thank you for submitting your manuscript entitled "Comparative functional genomics identifies unique molecular features of EPSCs" to Life Science Alliance. We invite you to submit a revised manuscript addressing the following points:

- Address Reviewer 1 specific point #4. Address specific points #1, #2 and #3 via discussion. Address minor points.
- Address Reviewer 2 concerns via discussion.

Thank you for this interesting contribution to Life Science Alliance. We are looking forward to receiving your revised manuscript.

Sincerely,

- A letter addressing the reviewers' comments point by point.
- An editable version of the final text (.DOC or .DOCX) is needed for copyediting (no PDFs).
- High-resolution figure, supplementary figure and video files uploaded as individual files: See our detailed guidelines for preparing your production-ready images, <https://www.life-science-alliance.org/authors>
- Summary blurb (enter in submission system): A short text summarizing in a single sentence the study (max. 200 characters including spaces). This text is used in conjunction with the titles of papers, hence should be informative and complementary to the title and running title. It should describe the context and significance of the findings for a general readership; it should be written in the present tense and refer to the work in the third person. Author names should not be mentioned.
- By submitting a revision, you attest that you are aware of our payment policies found here: <https://www.life-science-alliance.org/copyright-license-fee>

B. MANUSCRIPT ORGANIZATION AND FORMATTING:

July 19, 2022

RE: Life Science Alliance Manuscript #LSA-2022-01608-TR

Dr. Jianlong Wang
Columbia University Medical Center
Medicine
650 W. 168th St
William Black Building, 8th Floor
New York, New York 10032

Dear Dr. Wang,

Thank you for submitting your revised manuscript entitled "Comparative functional genomics identifies unique molecular features of EPSCs". We would be happy to publish your paper in Life Science Alliance pending final revisions necessary to meet our formatting guidelines.

- please upload your supplementary figures as single files and add your supplementary figure legends to the main manuscript text
- please incorporate the Supplemental References into the main Reference list
- the Supplemental Reagents list should be labeled and provided as a Table. Supplemental Table would be fine for this
- please add the Twitter handle of your host institute/organization as well as your own or/and one of the authors in our system
- please make accession GSE201305 publicly available at this point

A. FINAL FILES:

B. MANUSCRIPT ORGANIZATION AND FORMATTING:

**Submission of a paper that does not conform to Life Science Alliance guidelines will delay the acceptance of your

manuscript.**

The license to publish form must be signed before your manuscript can be sent to production. A link to the electronic license to publish form will be sent to the corresponding author only. Please take a moment to check your funder requirements.

Sincerely,

July 20, 2022

RE: Life Science Alliance Manuscript #LSA-2022-01608-TRR

Dr. Jianlong Wang
Columbia University Medical Center
Medicine
650 W. 168th St
William Black Building, 8th Floor
New York, New York 10032

Dear Dr. Wang,

Thank you for submitting your Research Article entitled "Comparative functional genomics identifies unique molecular features of EPSCs". It is a pleasure to let you know that your manuscript is now accepted for publication in Life Science Alliance. Congratulations on this interesting work.

DISTRIBUTION OF MATERIALS:

Again, congratulations on a very nice paper. I hope you found the review process to be constructive and are pleased with how the manuscript was handled editorially. We look forward to future exciting submissions from your lab.

Sincerely,
